# Belatacept as a Treatment Option in Patients with Severe BK Polyomavirus Infection and High Immunological Risk—Walking a Tightrope between Viral Control and Prevention of Rejection

**DOI:** 10.3390/v14051005

**Published:** 2022-05-09

**Authors:** Ulrich Jehn, Sami Siam, Vanessa Wiening, Hermann Pavenstädt, Stefan Reuter

**Affiliations:** Department of Medicine D, Division of General Internal Medicine, Nephrology and Rheumatology, University Hospital of Münster, 48149 Münster, Germany; sami.siam@ukmuenster.de (S.S.); vanessa.wiening@ukmuenster.de (V.W.); hermann.pavenstaedt@ukmuenster.de (H.P.); stefan.reuter@ukmuenster.de (S.R.)

**Keywords:** BK polyomavirus, BKPyV-associated nephropathy, kidney transplantation, immunosuppression, belatacept, allograft rejection

## Abstract

Balancing the immune system with immunosuppressive treatment is essential in kidney transplant recipients to avoid allograft rejection on the one hand and infectious complications on the other. BK polyomavirus nephropathy (BKPyVAN) is a viral complication that seriously threatens kidney allograft survival. Therefore, the main treatment strategy is to reduce immunosuppression, but this is associated with an increased rejection risk. Belatacept is an immunosuppressant that acts by blocking the CD80/86-CD28 co-stimulatory pathway of effector T-cells with marked effects on the humoral response. However, when compared with calcineurin-inhibitors (CNI), the cellular rejection rate is higher. With this in mind, we hypothesized that belatacept could be used as rescue therapy in severely BKPyV-affected patients with high immunological risk. We present three cases of patients with BKPyVAN-associated complications and donor-specific antibodies (DSA) and one patient who developed T-cell-mediated rejection after a reduction in immunosuppression in response to BKPyVAN. Patients were switched to a belatacept-based immunosuppressive regimen and showed significantly improved viral control and stabilized graft function. The cases presented here suggest that belatacept is a potential treatment option in the complicated situation of refractory BKPyV infection in patients with high immunological risk.

## 1. Introduction

BK virus (BKPyV) is a double-stranded DNA virus that belongs to the family Polyomaviridae [1,2]. In non-immunocompromised individuals, primary BKPyV infection occurs predominantly before adolescence, with an IgG seroprevalence of 87% in people aged 20–29 years, and it is mainly asymptomatic. By so-far-unknown mechanisms, viral persistence occurs after primary infection [3]. Under conditions of immunosuppression, which are necessary after allogenic organ transplantation, reactivation of BKPyV with enhanced viral replication might lead to severe complications and is a serious source of morbidity [2,4,5]. BKPyV-associated nephropathy (BKPyVAN) is a serious complication after kidney transplantation (KTx) that occurs in 1–10% of renal allograft recipients and endangers kidney allograft function and survival. Long-term outcomes of BKPyVAN are poor, with an allograft loss of approximately 90% if measures to modify immunosuppression are not taken [2,6].

Until now, there has been no specific antiviral treatment for BKPyV. Hence, reduction in immunosuppression is the cornerstone of the treatment strategy used against severe BKPyV infection/reactivation [7]. However, it appears that the use of everolimus (EVR), the mechanistic target of rapamycin (mTOR)-inhibitor, instead of mycophenolate as an immunosuppressant in patients with BKPyVAN offers favorable allograft outcomes, which is partly explained by the antiviral effect of mTOR-inhibitors [8,9]. In addition, it has been observed that BKPyVAN incidence is lower in EVR-based immunosuppressive regimens when compared with CNI-based regimens [10,11].

Belatacept is a CTLA-4-Ig chimeric fusion protein that was introduced in 2011. It inhibits a co-stimulatory pathway of effector T-cells by specifically binding to CD80/86, thereby blocking the interaction of CD80/86 with CD28, which activates effector T-cells [12].

In a post hoc analysis of BENEFIT and BENEFIT-EXT studies, belatacept was found to be superior in preventing the formation of de novo donor-specific antibodies (dnDSA) at 3 and 7 years after KTx when compared with cyclosporine A (CsA) [13]. In contrast, a cellular immune response might not be as strongly suppressed with belatacept, as evidenced by the increased risk of TCMR [14,15]. BKPyV data after KTx in patients treated with belatacept are rare. However, in [16], infection rates did not increase with de novo use of belatacept or after switching from calcineurin inhibitors (CNI) to belatacept when compared with using CsA, although overall infection rates were not high in these studies.

Almost nothing is known about the application of belatacept in the context of active viral complications after KTx, particularly BKPyVAN or significant DNAemia. In stable KTx patients, infectious complications have been found to be equally frequent in those receiving CNI when compared with those receiving belatacept [17]. There is no evidence-based therapeutic strategy for cases of BKPyV infection or BKPyVAN in patients treated with belatacept. In their review, Terrec et al. did not recommend discontinuing belatacept in these situations [16].

Here, we present three cases of refractory BKPyVAN and one case of refractory BKPyV DNAemia that were treated by converting their immunosuppressive therapy to a belatacept-based regimen as a rescue approach.

## 2. Case Presentations


**Case 1:**


The first case was a 58-year-old male patient who received an ABO-incompatible living-donor transplant after desensitization with rituximab and immunoadsorption with semi-selective devices. Induction therapy was performed with anti-thymocyte globulins (ATG), and initial immunosuppression consisted of immediate-release tacrolimus (Tac) (trough level 6–8 ng/mL), mycophenolic acid (MPA), and prednisone.

Four months after KTx, BKPyV viremia was initially diagnosed with 132,000 copies/mL. DSA-diagnostic was negative. A transplant biopsy showed BKPyVAN. Therefore, the immunosuppressive regimen was switched to CsA, EVR, and prednisone. Furthermore, intravenous immunoglobulins (IVIG, 0.5 g/kg body weight) were administered monthly nine times.

Four months later, four de novo donor-specific antibodies (dnDSA)—namely anti-HLA-A2 (mMFI~3.100FLU), anti-HLA-A68 (mMFI~700FLU), anti-HLA-DR7 (MFI~3.200FLU), and anti-HLA-DR53 (mMFI~3.100FLU)—were found. Based on the dnDSA, another renal biopsy was performed, which excluded anti-donor antibody-mediated rejection (ABMR) but again demonstrated BKPyVAN. Due to BKPyVAN, the estimated glomerular filtration rate (eGFR) had decreased from the initial baseline of 60–65 mL/min/1.73 m^2^ to around 40–45 mL/min/1.73 m^2^ (CKD-EPI formula) (Figure 1, blue line). A few weeks later, against the background of multiple Class I and II dnDSA, we decided to replace CsA with belatacept, so the patient was immunosuppressed with belatacept, EVR (target trough 3–5 ng/mL), and prednisone. One year later, three DSA (anti-HLA-DR7 (MFI~1000FLU), anti-HLA-DR53 (mMFI~700FLU) and anti-HLA-DQ5 (mMFI~600FLU)) were still detectable, but their levels had decreased.

After the conversion to belatacept, the patient’s kidney function remained stable with an eGFR around 40–45 mL/min/1.73 m^2^ (CKD-EPI formula) (Figure 1, blue line). BKPyV copies in plasma dropped from values of >100,000 copies/mL to a steady state of ~1000 copies/mL (Figure 2, blue line) one year after conversion.


**Case 2:**


This case involved a 63-year-old male patient who had received a postmortal donor transplant five years earlier. His initial immunosuppressive regimen consisted of Tac, MPA, and prednisone after induction therapy with basiliximab. BKPyVAN was biopsy-proven four months after KTx, and a decrease in allograft function and BKPyV DNAemia were observed. The diagnosis of BKPyVAN led to a conversion of the immunosuppressive triple-regimen to CsA (target trough 70–90 ng/mL), EVR (target trough 4–5 ng/mL), and prednisone. Furthermore, IVIG (at 0.5 g/kg of body weight) was administered monthly four times. BKPyV replication was controlled under this regimen with a reduction in BKPyV copies from 303,000/mL at the diagnosis of BKPyVAN to 100 copies/mL three months later. Nevertheless, allograft function deteriorated further. A second allograft biopsy at that time revealed acute TCMR 1A. Despite the use of steroid pulse therapy and the replacement of CsA with Tac (trough level 5–7 ng/mL), another biopsy three weeks later again showed ongoing TCMR 1A. Rejection was treated with another steroid pulse and with ATG (at a cumulative dose of 6 mg/kg of body weight).

As a last resort therapy in the dilemma of concurrent TCMR 1A and recent BKPyVAN, we switched the immunosuppressive regimen to belatacept, EVR, and prednisone. Subsequently, kidney function improved from a pre-terminal stage with an eGFR of 15–20 mL/min/1.73 m^2^ to an eGFR of 25–30 mL/min/1.71 m^2^, where it consolidated after a follow-up period of 3.5 years (Figure 1, green line). BKV replication remained stable at a low level of about 100 copies/mL without further deterioration (Figure 2).


**Case 3:**


Our third case was an 81-year-old male patient in good general condition who received KTx after receiving a postmortal donation. The initial immunosuppressive regimen consisted of Tac, MMF, and prednisone. Three months after KTx, a dnDSA (anti-DRB1, MFI~680FLU) was diagnosed. Two months later, co-replication of BKV (7400 Cop/mL) and CMV (750 copies/mL) occurred. Consequently, MPA was replaced with EVR. BKPyVAN was excluded by a kidney biopsy because the renal function had deteriorated from a baseline eGFR of 50–60 mL/min/1.73 m^2^ to an eGFR of 30 mL/min/1.73 m^2^. The biopsy showed nephrocalcinosis and vacuolization of 10% of the proximal tubular epithelial cells, which was interpreted as CNI-induced nephrotoxicity.

In this constellation of dnDSA, BKPyV DNAemia, and CNI toxicity, we decided to replace Tac with belatacept at a viral BKV load of 71,000 Cop/mL. After an initial further increase in BKPyV replication to a maximum of 108,900 cop/mL, BKPyV DNAemia decreased over the following few months to values of around 100 Cop/mL (Figure 2, yellow line). Although renal function did not improve with belatacept, it stabilized at ~30 mL/min/1.73 m^2^ (Figure 1, yellow line). A subsequent DSA diagnostic under belatacept one year later revealed a constant anti-DRB1 level with MFI~900FLU.


**Case 4:**


The fourth case was a 56-year-old male patient who received an ABO-incompatible living-donor transplantation from his wife. Thirteen months after KTx, BKPyVAN was diagnosed. BKPyV DNAemia was detected only four months after KTx, with an initial viral load of 14,690 copies/mL. After basiliximab induction and initial immunosuppression with Tac/MPA/prednisone, the regimen was switched to CsA/EVR/prednisone. Additionally, IVIG (0.5 g/kg body weight) was infused monthly four times. One year after KTx, dnDSA anti-HLA-B8 (MFI~800FLU) and -DR14 (MFI~800FLU) were diagnosed. A transplant biopsy revealed BKPyVAN without signs of rejection. Approximately one year later, CsA was replaced by belatacept due to progressive loss of renal function from a baseline eGFR of ~60 mL/min/1.73 m^2^ to an eGFR of ~35 mL/min/1.73 m^2^ to prevent antibody-mediated rejection in this immunologic high-risk situation. Under belatacept, renal function increased significantly to an eGFR of between 50–55 mL/min/1.73 m^2^ (Figure 1, red line). BKPyV replication eventually consolidated at between 500 and 2000 copies/mL (Figure 2, red line). DSA diagnostics were already negative at the time of conversion and remained so during the follow-up.

The most relevant demographic and clinical characteristics of the four presented patients are summarized in Table 1.

Because of the monthly belatacept administrations at our outpatient unit, follow-up periods for the four presented patients were similar. In Cases 1 and 4, which were ABO-incompatible living donor KTxs, induction therapy was performed with Rituximab and ATG. In Cases 2 and 3, basiliximab and ATG, respectively, were solely applied (Table 1). Belatacept was administered in doses of 5 mg/kg of body weight every 28 days. The first six doses at conversion were administered on days 1, 5, 15, 29, 43, and 57 according to a protocol published by Rostaing et al. [18].

BKPyV diagnostics were performed monthly within the first 6 months after KTx, every second month during months 6–12, and on indication. EDTA-plasma was used for the kPCR 150 PLX^®^ BKV-assay to quantify BKPyV DNA. The HLA-Antibody screening was performed three months after KTx and every 12 months after KTx as well as on indication. Biopsy specimens were examined histologically by a nephropathologist and discussed at the weekly pathology-nephrology conference.

## 3. Discussion

The four cases presented here demonstrate that belatacept can be successfully used in patients with concurrent BKPyV infection and immunological complications in the form of dnDSA or TCMR 1A and therefore might represent a valuable immunosuppressive option in this dilemma. However, we expressly point out that the immunosuppressive drug combinations used in the presented patients are off-label therapies.

In all presented patients, we observed good control of BKPyVAN or DNAemia and stabilization or improvement of renal function with belatacept. None of the four patients experienced rejection episodes during follow-up, which ranged from 12 months to 3.5 years.

It is postulated that the risk of antibody-mediated rejection is lower with belatacept when compared with a CNI-based regimen. Nevertheless, there is a concomitant increased risk of T-cell-mediated rejection [14,15]. In patients with a stable renal function, conversion from a CNI-based regimen to belatacept resulted in a higher biopsy-proven rejection rate but a lower incidence of dnDSA [17].

In contrast to the other three presented cases, Case 2 had experienced a recent steroid-refractory TCMR 1A. Because sufficiently strong immunosuppression was required without reactivation of BKPyVAN, we struggled to switch to belatacept and considered it as an ultima ratio therapy because of the increased risk of TCMR associated with belatacept. Nevertheless, we did not observe any recurrence of TCMR or BKPyVAN after switching to belatacept at a follow-up more than three years later.

The available data on viral infections in patients on belatacept are somewhat contradictory. Chavarot et al. describe a sevenfold increase in the incidence of CMV disease and an atypical, aggressive disease phenotype in their patients [19]. In response to that study, Kleiboeker et al. presented the case of a 56-year-old male patient who lost his CMV-specific cell-mediated immunity after conversion to belatacept, and they hypothesized that this was a possible reason for the observations by Chavarot et al. [20]. To note, in contrast to our patients, who received belatacept at a dose of 5 mg/kg of body weight, in the case presented by Kleiboeker et al. belatacept was dosed at 10 mg/kg of body weight [20]. However, in the study by Chavarot et al. belatacept was administrated at 5 mg/kg of body weight [19].

In contrast to these observations, BENEFIT and BENEFIT-EXT studies report similar results with belatacept and CsA for BKPyV and CMV-associated complications in patients [14,15]. These results were supported by a study by Bassil et al. that compared CMV, EBV, BKV, and JCV complications in de novo KTx patients with belatacept and CNI. They detected statistically significant differences only for EBV infections, which occurred more frequently with belatacept therapy [21]. However, in their recently published review of infectious complications in KTx patients treated with belatacept, Terrec et al. clearly state that the safety profile of belatacept for BKPyV DNAemia and BKPyVAN is relatively good and that their data do not show an increased risk of BKPyV infections when compared with CNI-based immunosuppressive therapy [16]. Unfortunately, to our knowledge, there is no study investigating the course of BKPyV DNAemia after conversion from a triple-immunosuppressive regimen consisting of CNI/MMF/prednisone or CNI/EVR/prednisone to a belatacept-based one.

The T-cell co-stimulation modulator, abatacept, which shares the mode of action with belatacept and is used in patients with rheumatic diseases, does not appear to increase the risk of opportunistic infectious complications when compared with placebo or non-biologic disease-modifying drugs [22].

It was evident that in addition to belatacept and steroids, which were 5 mg of prednisone in each of our four patients, a third immunosuppressive drug played an important role in both viral control and effective immunosuppression for preventing alloimmunity. In the patients presented herein, that drug was the mTOR-Inhibitor EVR with a target trough level of 3–5 ng/mL, rather than MPA, which is routinely used with belatacept. EVR was applied due to its antiviral potency, which has been proven to have comparable efficacy for BKPyVAN [8] and MPA/TAC in kidney transplant patients [10].

Immunosuppression with belatacept is associated with an increased risk of the EBV-related complication of post-transplant-lymphoproliferative disease (PTLD), particularly in patients who are seronegative for EBV-IgG [23], because the proliferation of EBV-infected B-cells cannot be controlled by EBV-specific CD4+ and CD8+ T-cells [24]. Mechanistically, Kühne et al. demonstrated that belatacept inhibits allo-specific de novo T-cell responses, whereas virus-specific memory T-cell responses are not inhibited because the virus-specific IL-2 response is not affected [25].

Weakened BKPyV-specific cellular immunity is presupposed for high-replicative BKPyV infection or reactivation [26]. Sufficient clearance of BKPyV DNAemia or BKPyVAN is linked to increasing BKPyV-specific T-cell response [27]. In contrast to immunocompetent individuals and KTx recipients with good BKV clearance after exposure, interferon γ cytokine release by CD4+ T-cells is usually reduced in KTx recipients. Kaur et al. concluded that both the quantity of CD4+ T-cells and their functionality strongly influence the antiviral response to BKPyV [26].

Therefore, one possible explanation for our observations regarding the successful use of belatacept in our patients is that virus-specific memory T-cells, which were likely to be responsible for the control of BKPyV DNAemia in the cases presented, had already formed before the conversion to belatacept or that belatacept had less impact on the functionality of these T-cells than CNI did.

We hypothesized that belatacept enhances the virus-specific immune response against BKPyV after conversion from the more unselective CNIs or MMF without significantly boosting allo-specific T-cell immunity. Simultaneously, humoral alloimmunity is known to be weaker under belatacept than under CNI therapy [28]. This is an important consideration in patients diagnosed with DSA. Interestingly, we observed persistent BKPyV DNAemia rather than complete viral clearance at low replication levels in all patients during follow-up.

A further step to elucidate the underlying mechanisms of BKPyV control under belatacept should be to test for BKPyV-specific T-cells before belatacept conversion and under ongoing therapy.

## 4. Conclusions

In conclusion, three of the four cases presented here indicate that conversion of immunosuppression to belatacept and EVR in patients at high risk of ABMR and concomitant BKPyVAN might be a valuable therapeutical option to treat BKPyV infection without exposing the allograft to ABMR. In addition, the findings from Case 2 suggest that a high immunological risk based on recent TCMR is not a mandatory contraindication to the use of belatacept to simultaneously maintain virus and rejection control in certain cases.

## Figures and Tables

**Figure 1 viruses-14-01005-f001:**
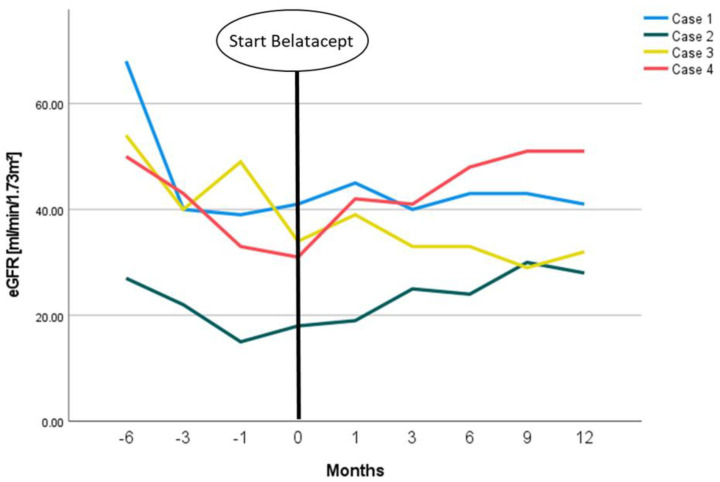
eGFR-courses of presented patients in relation to the start of belatacept treatment.

**Figure 2 viruses-14-01005-f002:**
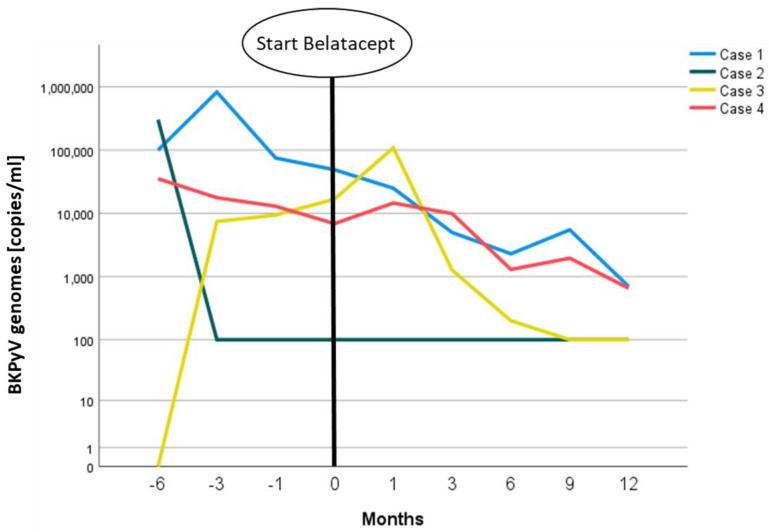
Courses of BKPyV replication in the blood plasma of presented patients in relation to the start of belatacept treatment measured by PCR.

**Table 1 viruses-14-01005-t001:** Patients’ demographic and clinical characteristics at belatacept conversion.

	Case 1	Case 2	Case 3	Case 4
**Sex**	male	male	male	male
**Age at start belatacept (years)**	57.3	59.4	80.5	52.9
**Time since KTx (months)**	8.5	8.0	7.0	11.8
**PRA %**	0	0	0	0
**Donation type**	ABO-I living donor KTx	postmortal	postmortal	ABO-I living donor KTx
**Time on dialysis before KTx (months)**	12	118	49	10
**Previous KTx**	none	none	none	none
**CMV mismatch**	D−/R+	D−/R−	D−/R+	D−/R+
**Induction therapy**	ATG + RTX	Basiliximab	ATG	ATG + RTX
**Initial immunosuppressive regimen**	Tac/MMF/Pred	Tac/MMF/Pred	Tac/MMF/Pred	Tac/MMF/Pred
**Diagnosis of ESRD**	Hypertensive nephropathy	Medullary sponge kidneys	Hypertensive nephropathy	IgA nephropathy

Abbreviations: KTx: kidney transplantation; HLA: human leukocyte antigen; PRA: panel reactive antibodies; ATG: anti-thymocyte globulins; RTX: Rituximab; Tac: Tacrolimus; MMF: Mycophenolate mofetil; Pred: Prednisone; CMV: Cytomegalovirus; ESRD: end-stage renal disease.

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
