# Peer review of "Belatacept as a Treatment Option in Patients with Severe BK Polyomavirus Infection and High Immunological Risk—Walking a Tightrope between Viral Control and Prevention of Rejection"

_viruses, 2022, doi:10.3390/v14051005_

Round 1
Reviewer 1 Report
I consider that the article presents very interesting data concerning the treatment of kidney transplant patients with refractory BKV infection. The relevant data of each case is displayed. However, there are a few aspects that required to be addressed:
Main comments:
- The introduction has to present information about the BK virus. I recommend adding a very short paragraph at the beginning about the BKV epidemiology including primary infection, latency, and then the rate of morbidity by BKV after kidney transplantation. For example:
BK virus (BKPyV) is a member of the Polyomaviridae family of double-stranded DNA viruses. Primary BKPyV infection is mainly asymptomatic and occurs predominantly before adolescence, after primary infection and, by a yet unknown mechanism, they establish viral persistence. Under conditions of immunosuppression, enhanced viral replication leads to severe diseases. Reactivation of the human polyomavirus BK (BKV) after kidney transplantation can be a serious source of morbidity. BKPyVAN is an important complication of renal transplant, occurring in 1-10% (?) of renal allograft recipients.
Recommended reading:
- Knowles, W. A. et al. Population-based study of antibody to the human polyomaviruses BKV and JCV and the siI simian polyomavirus SV40. J. Med. Virol. 71, 115–123 (2003).
- Egli, A. et al. Prevalence of polyomavirus BK and JC infection and replication in 400 healthy blood donors. J. Infect. Dis. 199, 837–846 (2009).
- Ginevri, F. et al. Polyomavirus BK infection in pediatric kidney-allograft recipients: A single-center analysis of incidence, risk factors, and novel therapeutic approaches. Transplantation 75, 1266–1270 (2003).
- Hirsch, H. H. & Randhawa, P. BK virus in solid organ transplant recipients. Am. J. Transplant. 9, S136–S146 (2009).
- Brennan, D. C. et al. Incidence of BK with tacrolimus versus cyclosporine and impact of preemptive immunosuppression reduction. Am. J. Transplant. 5, 582–594 (2005).
- In case 2. The BKPyV DNA levels changed from 330.000 copies/ml to 100 copies, this is not shown in the graph (there is something wrong). It is no mention of figure 2 in case 2.
- In case 3. The BKPyV DNA levels in the text are not reflected in the graph, In the graph, they start at 0 and finish at 100 copies/ml.
- In line 144 table 1. It is not properly introduced, the relevance or importance of the data presented in the table has to be presented in the text.
- ATG therapy (line 144) is not defined in the text, it is the same as antithymocyte globulin, is it the same as thymoglobulin (Line 65) please use only one term if referring to the same therapy and describe the abbreviation in the text when first used.
- Figure 2. In the y ax: BKV genomes (copies/ml).
- Please put attention to the introduction of the abbreviations in the text, there are not introduced when first mentioned but later or they are not explained in the text.
Minor comments:
Line 11 BK-Polyomavirus nephropathy--- remove line and capitals BK polyomavirus nephropathy (BKVN)
- Line 15-calcineurin-inhibitors –-- introduce abbreviation (CNI)
- Line 17 BKV--- explain the abbreviation BK polyomavirus
- Line 18- donor-specific antibodies--- introduce abbreviation (DSA)
- Line 18 T-cell-mediated rejection--- introduce abbreviation (TCMR)
- Line 37 BKV(N)- replace by BKVN
- Line 47-BKN was not defined -- BK polyomavirus (BKV)
- Line 52- DNAemia, could you please define it better in the text, for example, the presence of viral genomes in the blood ( BKPyV DNAemia).
- Line 68 and in all the text, replace Cop/m with copies/ml
- Line 75 ABMR- include abbreviation (anti-donor antibody-mediated rejection)
- Line 76 could you please introduce the explanation for eGFR (estimated glomerular filtration rate)
- Line 98 this was defined previously: - T-cell mediated rejection so only use the abbreviation (TCMR 1A)
- Line 127 BKDNAemia space missing BK DNAemia
- AB0 or ABO? For example line 126 vs line 143
- Line 144 and 145 abbreviation NTx not defined
- Line 150 missing some explanation of the kit (in red suggested change).. .. EDTA-blood was used for quantification of BKV DNA using kPCR 150 PLX® BKV-assay.
- Keywords: BK Polyomavirus- BK polyomavirus (not capital in polyoma)
Reviewer 2 Report
Very interesting immunosuppressive treatment approach in a difficult situation.
In table 1 it lacks time unit for “since KTx “
185 I don’t understand « ultima ratio therapy »
